# Geographic and Climatic Attributions of Autumn Land Surface Phenology Spatial Patterns in the Temperate Deciduous Broadleaf Forest of China

**Weiguang Lang [1], Xiaoqiu Chen [1,\*] , Liang Liang [2], Shilong Ren [1]  and Siwei Qian [1]**

[1]  Laboratory for Earth Surface Processes of the Ministry of Education, College of Urban and Environmental Sciences, Peking University, Beijing 100871, China

[2]  Department of Geography, University of Kentucky, Lexington, KY 40506, USA

\*  Correspondence: cxq@pku.edu.cn; Tel.: +86-10-62753976

**Abstract:** Autumn vegetation phenology plays a critical role in identifying the end of the growing season and its response to climate change. Using the six vegetation indices retrieved from moderate resolution imaging spectroradiometer data, we extracted an end date of the growing season (EOS) in the temperate deciduous broadleaf forest (TDBF) area of China. Then, we validated EOS with the ground-observed leaf fall date (LF) of dominant tree species at 27 sites and selected the best vegetation index. Moreover, we analyzed the spatial pattern of EOS based on the best vegetation index and its dependency on geo-location indicators and seasonal temperature/precipitation. Results show that the plant senescence reflectance index-based EOS agrees most closely with LF. Multi-year averaged EOS display latitudinal, longitudinal and altitudinal gradients. The altitudinal sensitivity of EOS became weaker from 2000 to 2012. Temperature-based spatial phenology modeling indicated that a 1 K spatial shift in seasonal mean temperature can cause a spatial shift of 2.4–3.6 days in EOS. The models explain between 54% and 73% of the variance in the EOS timing. However, the influence of seasonal precipitation on spatial variations of EOS was much weaker. Thus, spatial temperature variation controls the spatial patterns of EOS in TDBF of China, and future temperature increase might lead to more uniform autumn phenology across elevations.

**Keywords:** vegetation index; end of the growing season; leaf fall date; spatial pattern; spatial phenology model; climatic control

## 1. Introduction

Tracking the vegetation growing season is crucial for assessing an ecosystem response to climate change [1] and its impacts on carbon, water and energy exchange between vegetation and the atmosphere [2–5]. Some studies show that autumn phenology may have a greater control than spring phenology over the growing season length [6] and net ecosystem production (NEP) [7]. However, current research is focused mainly on understanding the relationship between autumn phenology and climate drivers [8,9], which have shown the importance of autumn temperatures in determining the timing of leaf senescence [10,11]. On the other hand, the spatial variation of autumn phenology and its climatic controls have received far less attention. Earlier attempts for identifying environmental controls of phenological spatial patterns can be traced to Hopkins' work. He proposed a "Bioclimatic Law" to estimate the offset in onset of spring as a function of latitude, longitude and elevation in the eastern U.S. [12]. Subsequently, multiple regression equations among average phenological dates and geo-location factors were constructed to estimate phenological gradients, along with latitude, longitude and elevation in different regions [13–16]. As geo-location factors are not climatic elements, they could

not explain the essential environmental causes of spatial difference of phenological occurrence dates or examine the spatial responses of phenological occurrence dates to climatic difference. To reveal climate drivers of phenological spatial patterns, Chen and Xu [5] analyzed relationships between the leaf fall end date of *Ulmus pumila* trees and the preseason temperature across geographic locations in China's temperate zone, and found that the spatial differences of preseason temperature control the spatial differences of *Ulmus pumila* leaf fall date. Whether this kind of spatial association based on an individual species is applicable to the ecosystem level is still unclear. Thus, a further examination of the spatial pattern of autumn phenology and its attributions at broad spatial scales over continuous vegetated land areas, such as through the use of remote sensing, is needed. Since spatial patterns of phenology reflect an adaptation of plant life cycle stages to different climates [17], modeling the spatial pattern and its interannual variation in autumn phenology will facilitate a better understanding of phenological responses to recent climate change.

Remotely sensed vegetation indices (VIs) have been commonly used to extract the start (SOS) and end (EOS) of the growing season [18–20]. However, few studies have assessed the reliability of different vegetation indices in extracting EOS. For deciduous broadleaf forests, some comparison studies indicated that the Normalized Difference Vegetation Index (NDVI) does not perform better than other VIs [21–24]. Meanwhile, a new vegetation index, the Plant Senescence Reflectance Index (PSRI), was proposed to determine the stage of leaf senescence and fruit ripening, as it is sensitive to carotenoid retention or accumulation in senescing leaves and ripening fruit [25]. The PSRI value increases during leaf senescence because of the increase in the ratio of carotenoids to chlorophylls. Spectral measurements in northern European boreal forests and moorland throughout the growing season showed that PSRI has similar sensitivity to vegetation dynamics with NDVI in spring, but higher sensitivity than NDVI in autumn [26,27]. Recently, Ren et al. firstly determined PSRI-derived SOS and EOS using moderate resolution imaging spectroradiometer data in the Inner Mongolian grassland, and found a similar performance of PSRI with NDVI [28]. More applications of PSRI should be made to evaluate its ability for identifying EOS in different vegetation types.

In this study, we first compared the spatial pattern of EOS dates retrieved from different VIs with a spatial pattern of ground-observed multispecies mean leaf fall dates at station/pixel scales, and then selected the best VI based on its performance in reflecting the ground reference. Then, we analyzed the relationships between EOS date and geo-location factors (such as latitude, longitude and altitude). Moreover, we constructed spatial phenology models to identify climatic controls of EOS spatial patterns in a multi-year average and for individual years.

## 2. Materials and Methods

### 2.1. Study Area

As the reliability of land surface phenological detection depends highly on selected vegetation types and their seasonality [29–32], our study focused on the temperate deciduous broadleaf forest (TDBF) with a distinct seasonal nature in northeastern China [33]. The existing TDBF is distributed mainly in mountainous areas, including the middle part of the Da Hinggan Ling Mountains, the Xiao Hinggan Ling Mountains, the Changbai Shan Mountains, the Yan Shan Mountains, the Taihang Shan Mountains and the Qin Ling Mountains from north to south with elevations ranging mainly from 200 to 2000 meters above sea level (Figure 1). The vegetation formations and sub-formations in the TDBF mainly contain Acer, Ulmus, Quercus, Robinia, Populus, Betula and Salix forest or woodland. The climate type in the TDBF is a temperate monsoonal climate with a hot and rainy summer and a cold and dry winter. The annual mean temperature increases from −8.9 °C in the north to 14.9 °C in the south, while the annual mean total precipitation ranges from 234.7 mm in the northern Yan Shan Mountains to 1456.5 mm in the Qin Ling Mountains. Because of the remarkable spatial heterogeneity of thermal-moisture conditions across the study areas, the plant phenology displays significant spatial

variations, which makes the TDBF a suitable biogeographical region for exploring spatial patterns of vegetation phenology and its climate drivers.

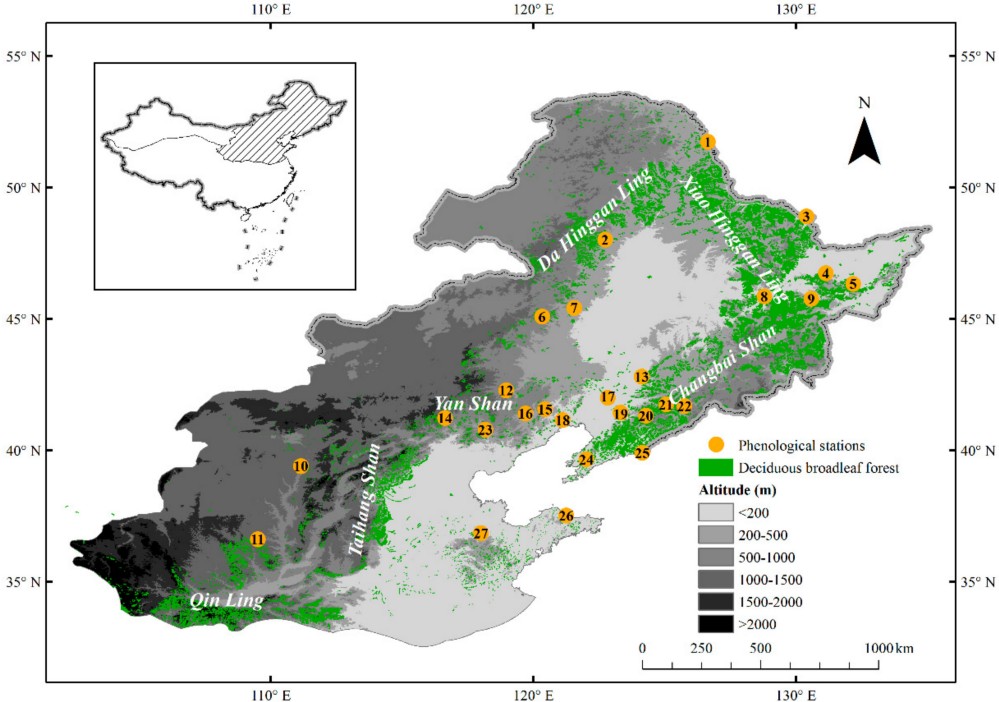

**Figure 1.** Distribution of temperate deciduous broadleaf forests and phenological observation stations in northeastern China.

## 2.2. Phenological and Meteorological Data

Ground-observed phenological data were acquired from the China Meteorological Administration (CMA). The CMA phenological observation network is the largest phenological observation system in China, and came into operation in 1980. At present, 446 stations regularly undertake phenological observations across the country, in which 27 stations are located within the TDBF area (Figure 1). According to the availability of phenological data and the representativeness of species, we selected the end of leaf fall stage of the following widely distributed species to validate satellite-retrieved leaf fall stage: *Populus berolinensis*, *Populus simonii*, *Populus X canadensis*, *Populus tomentosa*, *Robinia pseudoacacia*, *Sophora japonica*, *Salix matsudana*, *Salix babylonica*, and *Ulmus pumila*. The geographical coordinates and elevation of phenological stations and selected plant species at each station are shown in Table S1.

Moderate resolution imaging spectroradiometer (MODIS) data have been available since 2000, however, phenological data beyond 2012 were not yet available at the time of this study. Thus, we utilized a ground-observed 13 year continuous record of the leaf fall end date of each species and MODIS data from 2000 to 2012. For each species, at least three individual plants were selected as fixed observation objects at a given station. The leaf fall end date was identified as the day when more than 90% of the leaves fell off in more than half of the observed trees [34]. We calculated the average leaf fall end date of all indicator species at each station to represent roughly the leaf fall end date at plant community scales, which should be more comparable with the remotely sensed EOS than the leaf fall end date of a single species.

Daily temperature and precipitation data from 2000 to 2012 were acquired from the China Meteorological Forcing dataset (CMFD). This dataset is a fusion of data from multiple sources, including the Global Land Data Assimilation System (GLDAS) forcing data, Princeton forcing data, Global Energy and Water Cycle Experiment—Surface Radiation Budget (GEWEX-SRB) downward shortwave radiation, the Tropical Rainfall Measuring Mission (TRMM) satellite precipitation analysis

data (3B42) and GLDAS precipitation and CMA station data [35]. This data product is available at a spatial resolution of 0.1 degrees and a temporal resolution of 3 h with seven basic climatic factors (e.g., near-surface air temperature, precipitation, maximum wind speed, etc.). In this study, daily mean air temperature was calculated by the average of the eight temperature values in each day, while daily precipitation was calculated by an arithmetic sum of the eight values of accumulated precipitation every 3 h within a day.

### 2.3. MODIS Data and Processing

We used Moderate Resolution Imaging Spectroradiometer (MODIS) Surface Reflectance 8-Day L3 Global 500 m product (MOD09A1, V5) distributed by the Land Processes Distributed Active Archive Center (LP DAAC). Surface reflectance was computed from seven MODIS Level 1B bands (band 1 to 7 centered at 648 nm, 858 nm, 470 nm, 555 nm, 1240 nm, 1640 nm and 2130 nm, respectively) after removal of the atmospheric effect [36], in which the surface reflectance data from bands 1–5 were used to calculate the six types of vegetation indices, including the Normalized Difference Vegetation Index (NDVI), Land Surface Water Index (LSWI), Enhanced Vegetation Index (EVI), Wide Dynamic Range Vegetation Index (WDRVI), Green-Red Ratio Index (GRVI) and Plant Senescence Reflectance Index (PSRI). Detailed descriptions of these indices are shown in Table 1.

**Table 1.** Descriptions of vegetation indices used in this study.

| Abbreviations [1] | Moderate Resolution Imaging Spectroradiometer (MODIS) Bands | Equations [2] | References |
|---|---|---|---|
| NDVI | 1,2 | $NDVI = (R_{NIR} - R_{RED})/(R_{NIR} + R_{RED})$ | [37] |
| LSWI | 2,5 | $LSWI = (R_{NIR} - R_{SWIR})/(R_{NIR} + R_{SWIR})$ | [38] |
| EVI | 1,2,3 | $EVI = 2.5 \times (R_{NIR} - R_{RED})/(1 + R_{NIR} + 6 \times R_{RED} - 7.5 \times R_{BLUE})$ | [39] |
| WDRVI | 1,2 | $WDRVI = (0.2 \times R_{NIR} - R_{RED})/(0.2 \times R_{NIR} + R_{RED})$ | [40] |
| GRVI | 1,4 | $GRVI = (R_{GREEN} - R_{RED})/(R_{GREEN} + R_{RED})$ | [41] |
| PSRI | 1,2,4 | $PSRI = (R_{RED} - R_{GREEN})/R_{NIR}$ | [25] |

[1] NDVI: Normalized Difference Vegetation Index, LSWI: Land Surface Water Index, EVI: Enhanced Vegetation Index, WDRVI: Wide Dynamic Range Vegetation Index, GRVI: Green-Red Ratio Index, PSRI: Plant Senescence Reflectance Index. [2] NIR and SWIR denote near infrared and shortwave infrared, respectively.

MOD09A1 provides a quality assurance (QA) layer (indicating the abnormal pixels contaminated by snow, ice or clouds on the data) and a day of the year (DOY) layer (marking the dates of data acquisition for respective pixels). Utilizing the QA and DOY flags, we identified the abnormal VI pixel values that were affected by snow, ice or clouds. Then, we replaced the abnormal VI values during the start and end stages of the year using the nearest normal VI values and other abnormal VI values with linearly interpolated VI values of the two nearest normal VI values at both sides. Moreover, we identified the date ($D_f$) when the minimum VI (maximum PSRI) occurred in the first half of the reconstructed VI sequence and the date ($D_s$) when the minimum VI (maximum PSRI) occurred in the second half of the reconstructed VI sequence. All VI values from 1 January to $D_f$ and from $D_s$ to 31 December were further replaced by the average of VI values corresponding to $D_f$ and $D_s$. Here, the minimum VI (maximum PSRI) value represents the initial background VI value [42]. Finally, we smoothed the VI time series with a five point moving average filter (Figure 2). As the double logistic method is demonstrated to be particularly useful for describing annual VI time series for phenological monitoring [31,43], we fitted the smoothed VI time series with the double logistic function [28,32]. The corresponding formula is as follows:

$$VI(t) = VI_{\min} + (VI_{\max} - VI_{\min}) \times (-1)^n \times \left( \frac{1}{1 + e^{I \times (S-t)}} + \frac{1}{1 + e^{D \times (E-t)}} - 1 \right) \qquad (1)$$

where *VI(t)* is the VI value of a given pixel at time *t*; $VI_{max}$ and $VI_{min}$ are the maximum and minimum VI values in each annual time series; *I* and *D* denote the maximum falling and rising slopes at the inflection points of each VI curve and *S* and *E* denote the corresponding dates of *I* and *D*. *n* equals to 1 for PSRI, while *n* equals to 0 for other VIs.

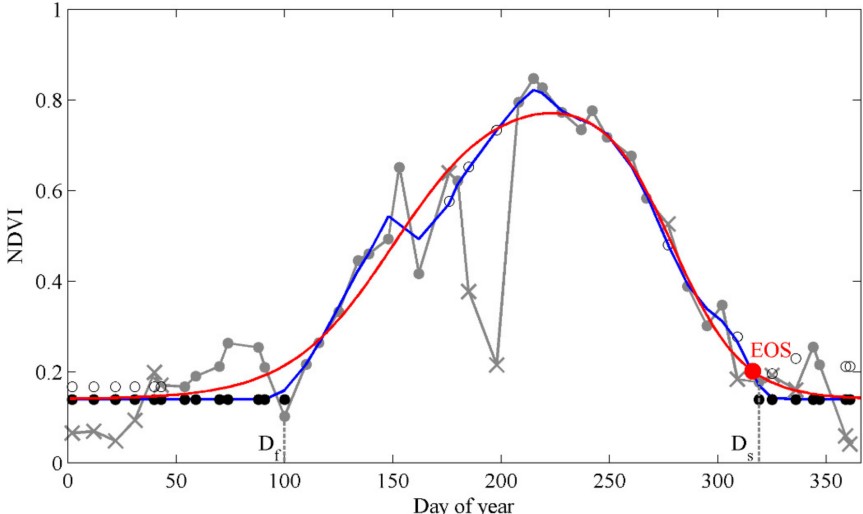

**Figure 2.** Schematic diagram for determining the end of the growing season taking the Normalized Difference Vegetation Index (NDVI) as an example. The grey line denotes the raw NDVI time series. The grey dots and crosses represent normal and snow/ice/cloud-contaminated abnormal NDVI values, respectively. The circles are the interpolated vegetation index (VI) values that replaced the abnormal values. $D_f$ and $D_s$ are the dates when the minimum VI (maximum Plant Senescence Reflectance Index (PSRI)) occurred in the first half of the reconstructed VI sequence and in the second half of the reconstructed VI sequence, respectively. The black dots denote the replaced VI values from 1 January to $D_f$ and from $D_s$ to 31 December. The blue line denotes the smoothed NDVI time series using the five point moving average. The red line is the fitted double logistic curve.

The EOS dates (day of year) were then identified using Equation (2) [32] at each pixel and for each year, as they are the dates of the last local minima curvature change rate of the fitted curves. All data processing was conducted with MATLAB software (Matlab R2014a, MathWorks Inc. Natick, MA, USA).

$$EOS = E + \frac{2 \times \ln\left(\sqrt{3} - \sqrt{2}\right)}{D} \tag{2}$$

To compare the ability of different VIs to capture the spatial pattern of ground autumn phenology, we calculated the root mean square error [RMSE; Equation (3)] and Pearson correlation coefficient (*r*) between the spatial series of the ground-observed LF date at 27 stations and the VI-derived EOS date at the corresponding pixels. The comparison was made both for individual years as well as for a multi-year average.

$$RMSE = \sqrt{\frac{\sum\limits_{i=1}^{n} (EOS_i - LF_i)^2}{n}} \tag{3}$$

where $EOS_i$ and $LF_i$ are the remotely-sensed end date of the growing season and the ground-observed leaf fall end date at station *i*, respectively. *n* is the number of stations.

The best performing VI was selected by the lowest RMSE value and highest correlation (*r*) for multi-year average, and by the largest number of years with the lowest RMSE value and highest correlation for individual years. Then, we used the best performing VI to identify the EOS for each pixel over the entire study area.

To ensure the accuracy of identified EOS dates, we eliminated pixels with the following attributes: (1) The EOS date did not appear in the current year; (2) The annual maximum VI (minimum PSRI) was not detected from April to October. The satellite-derived EOS dates were resampled to a spatial resolution of 0.1 degrees using a moving window averaging method to match the spatial resolution of climatic data. Because of the removal of abnormal pixels, the total number of pixels in TDBF were different in different years. To ensure the same geographic range among the different years, we extracted the common pixels from 2000 to 2012, and obtained 4091 pixels for further study.

### 2.4. Spatial Phenology Models Based on Geo-Location Indicators and Climatic Factors

To examine the attribution of yearly and multi-year average EOS spatial patterns, multiple regression models were created among the EOS date and geo-location indicators (latitude, longitude and elevation) across the 4091 common pixels [16]. The elevations at each pixel were determined by Digital Elevation Model (DEM) data extracted from the Global 30 Arc-Second Elevation (GTOPO30) dataset, in which DEM data were resampled from the spatial resolution of 30″ to 0.1° using a cubic convolution method in ArcGIS 10.3 software. The multiple regression model is shown as follows:

$$E\,O\,S(x, y, z) = a_0 + a_x \times x + a_y \times y + a_z \times z \tag{4}$$

where *EOS (x, y, z)* is the EOS date (in day of year) at latitude *x*, longitude *y* and altitude *z*; $a_0$ is a constant; $a_x$, $a_y$ and $a_z$ are regression coefficients denoting latitudinal sensitivity, longitudinal sensitivity and altitudinal sensitivity, respectively. Partial correlation analysis was also used to detect dependency of the EOS spatial pattern on each of the geo-location indicators (*x*, *y* and *z*), respectively.

We used mean air temperature and accumulated precipitation spatial series during the optimum length period to correlate with the EOS spatial series. The basic hypothesis was that the station-to-station variation of a phenological event occurrence date over an area is mainly influenced by station-to-station variation of climatic factors within a particular length period (LP) of days during and before its occurrence over the area [5]. The LP in this study was determined as the time periods during which the pixel-to-pixel variation in daily mean temperature or accumulated precipitation affects the pixel-to-pixel variation of EOS most remarkably in a year. The computation process was implemented through the following steps: First, we defined the number of days between the earliest and latest dates of the EOS spatial series across all the pixels in a given year as the basic length period (bLP). Then, we calculated backwards (to an earlier time direction) the mean air temperature and accumulated precipitation spatial series across all the pixels in the year during bLP + mLP. Here, mLP was a moving length period prior to the earliest date of EOS spatial series, which moved with a step length of 1 day, namely, during the bLP+1 day, bLP+2 day, bLP+3 day, etc. The maximum mLP was set to 60 days, as the mean temperature and accumulated precipitation prior to this period were usually assumed to have limited effects on spatial variation of occurrence dates of a phenological event. Thus, the LP is defined as follows:

$$LP = bLP + mLP \tag{5}$$

Furthermore, we calculated the correlation coefficients between the EOS spatial series and mean temperature/accumulated precipitation spatial series during different testing LPs across all the pixels in a specific year. Finally, we obtained the optimum LP based on the highest correlation between the EOS spatial series and the corresponding mean temperature/accumulated precipitation spatial series. The temperature or precipitation within the optimum LP across all the pixels in the year was thought of as the major influence factor of EOS spatial variation. Hence, the optimal temperature/precipitation-based spatial phenology model for each year can be created as:

$$EOS = a + b \times X \tag{6}$$

where *X* is the mean temperature or accumulated precipitation during the optimum LP; *a* and *b* are the intercept and slope of the equation, respectively. To evaluate the relative importance of temperature and precipitation on controlling EOS spatial patterns, we further executed a partial correlation analysis between EOS spatial series and temperature/precipitation spatial series during the optimum LP. The significance levels of partial correlation were evaluated based on two-tailed significance tests.

## 3. Results

### 3.1. Ground Validation of EOS Retrieved from Different Vegetation Indices

We used the spatial RMSE to measure the station-to-station differences between satellite-derived EOS dates and ground-observed LF dates in each year. Results show that the medians and ranges of RMSE for EOSs retrieved from all six VIs were between 14 days and 23.2 days, and between 5.6 days and 11 days, respectively, in which either the median or range of RMSE for PSRI-derived EOS was the smallest (Figure 3).

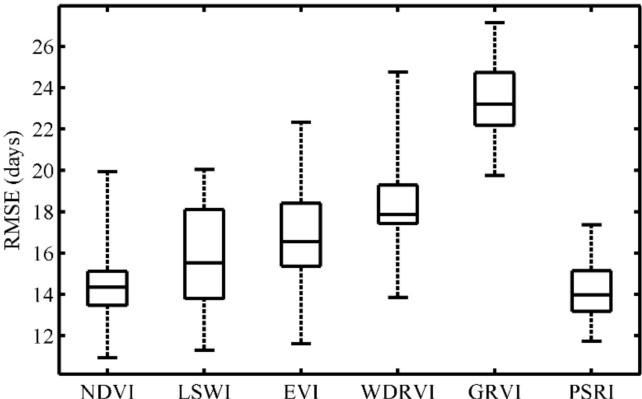

**Figure 3.** Medians and ranges of root mean square error (RMSE) between the growing season end dates retrieved from different vegetation indices and leaf fall end dates from 2000 to 2012.

To evaluate the spatial variation consistency between satellite-derived EOS dates and ground-observed LF dates, we conducted a correlation analysis between EOS spatial series and LF spatial series across all stations/pixels on a multi-year mean and for each year. On average, EOS dates retrieved from all six VIs were significantly ($p < 0.001$) and positively correlated with LF dates, although the satellite estimates were systematically earlier (Figure 4). In addition, relatively high correlations ($r \geq 0.7$) were found between all EOS estimates and LF. The correlation coefficient between the PSRI-derived EOS and LF was also the highest (0.76; Figure 4f), followed by that of LSWI (0.74; Figure 4b).

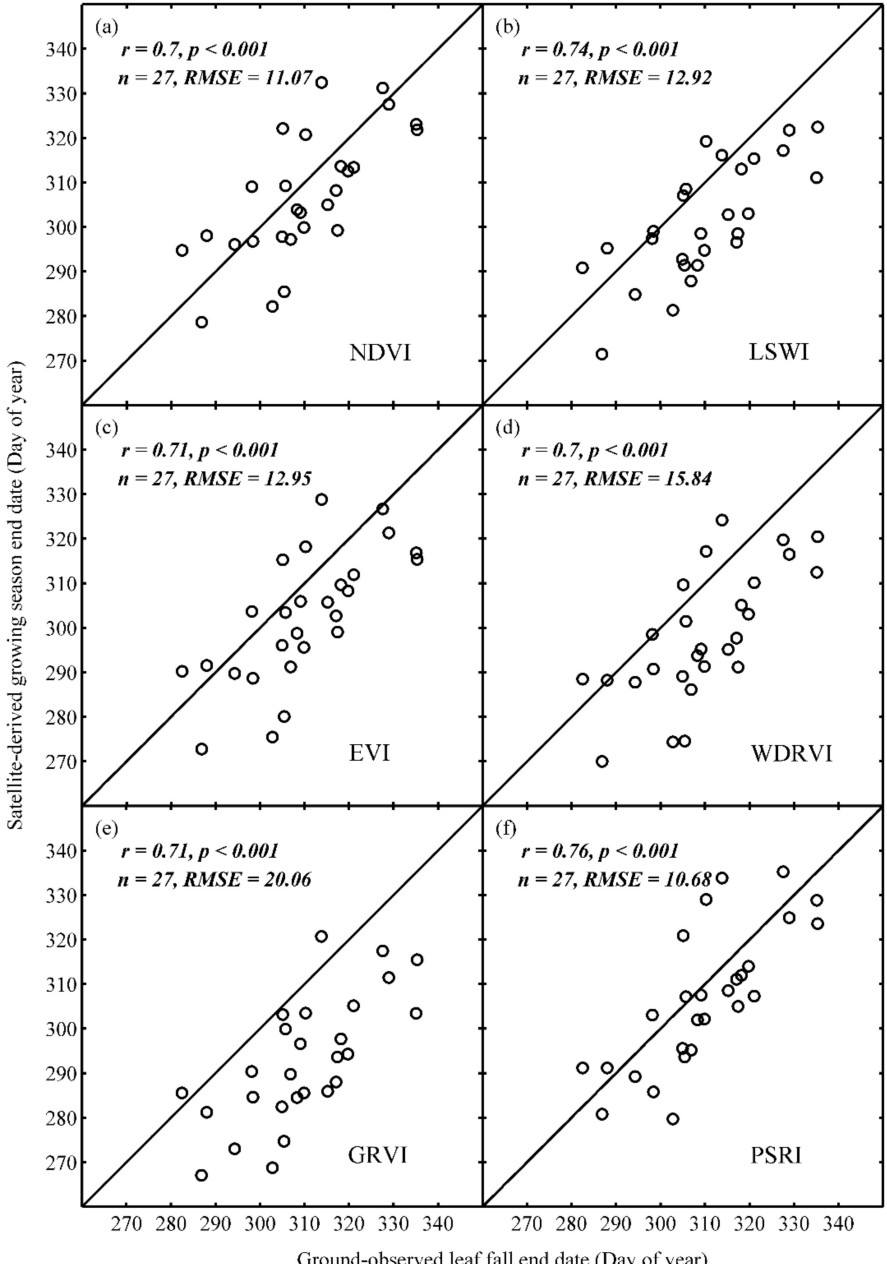

**Figure 4.** Correlation coefficients and errors (RMSE) in spatial series between multi-year mean growing season end dates retrieved from different VIs and multi-year mean ground-observed leaf fall end dates over 27 stations/pixels.

For each year, the correlation analysis showed a consistently positive relationship. The number of years with a significantly positive correlation ($p < 0.01$) ranged from 7 years (GRVI) to 12 years (PSRI) out of the 13 years in total (Table 2). The PSRI-based EOS performed better than other VIs in all cases, in terms of yielding relatively higher correlation coefficients.

In summary, an outstanding performance of the PSRI-derived EOS was demonstrated by its consistence with ground-observed LF. In contrast, the performance of the GRVI-derived EOS appeared to be the poorest. Performances of EOS retrieved from the other four VIs fall in between these two. Among the four VIs, the NDVI-derived EOS was the most accurate (with the smallest RMSE; Figure 4a) and the LSWI-derived EOS could more consistently reflect the spatial variation of ground-observed LF (with largest correlation coefficient; Figure 4b). Finally, we selected the PSRI-derived EOS for the further analysis.

**Table 2.** Correlation coefficients in spatial series between growing season end dates retrieved from different VIs and ground-observed leaf fall end dates in each year.

| Year | NDVI | LSWI | EVI | WDRVI | GRVI | PSRI |
|------|------|------|------|-------|------|------|
| 2000 | 0.76 ***[1] | 0.82 *** | 0.71 *** | 0.76 *** | 0.68 *** | 0.75 *** |
| 2001 | 0.59 ** | 0.57 ** | 0.57 ** | 0.62 *** | 0.55 ** | 0.62 *** |
| 2002 | 0.59 ** | 0.73 *** | 0.53 ** | 0.63 *** | 0.49 * | 0.53 ** |
| 2003 | 0.64 *** | 0.69 *** | 0.66 *** | 0.66 *** | 0.36 | 0.70 *** |
| 2004 | 0.55 ** | 0.63 *** | 0.56 ** | 0.59 ** | 0.65 *** | 0.64 *** |
| 2005 | 0.47 * | 0.61 *** | 0.49 ** | 0.49 ** | 0.56 ** | 0.64 *** |
| 2006 | 0.56 ** | 0.70 *** | 0.68 *** | 0.66 *** | 0.63 *** | 0.67 *** |
| 2007 | 0.31 | 0.44 * | 0.42 * | 0.37 | 0.43 * | 0.55 ** |
| 2008 | 0.35 | 0.40 * | 0.35 | 0.32 | 0.37 | 0.47 * |
| 2009 | 0.51 ** | 0.55 ** | 0.56 ** | 0.44 * | 0.47 * | 0.67 *** |
| 2010 | 0.55 ** | 0.65 *** | 0.57 ** | 0.52 ** | 0.66 *** | 0.75 *** |
| 2011 | 0.79 *** | 0.59 ** | 0.79 *** | 0.78 *** | 0.52 ** | 0.71 *** |
| 2012 | 0.57 ** | 0.54 ** | 0.18 | 0.47 ** | 0.45 * | 0.69 *** |

[1] * $p < 0.05$, ** $p < 0.01$, *** $p < 0.001$.

### 3.2. Spatial Patterns of EOS and Their Relation to Geo-Location Factors

Based on PSRI data from 2000 to 2012 at each pixel, we extracted the EOS over the entire TDBF area. The multi-year mean EOS ranged from 272 DOY to 346 DOY, and occurred earlier at higher latitudes than at lower latitudes. Specifically, the earliest EOS was detected at the northern parts of the study area, namely in the Da Hinggan Ling Mountains and Xiao Hinggan Ling Mountains, whereas the latest EOS was found in the south, such as in the Qin Ling Mountains and Taihang Shan Mountains (Figure 5).

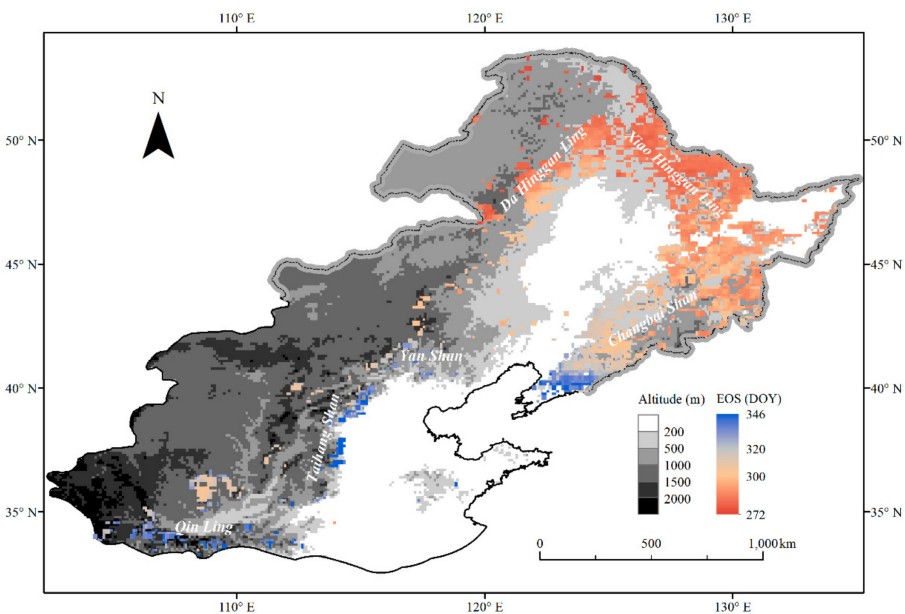

**Figure 5.** Spatial pattern in multi-year mean PSRI-derived growing season end dates in the temperate deciduous broadleaf forest area of northeastern China from 2000 to 2012.

Multiple regression equation slopes among multi-year mean EOS and geo-location indicators represented a latitudinal sensitivity of −2.98 days per degree northward, a longitudinal sensitivity of −1.03 days per degree eastward and an altitudinal sensitivity of −1.5 days per 100 m upward. This multiple regression model explained 80% of the spatial variance of multi-year mean EOS, with a simulation error of 7.1 days. Partial correlation analyses showed significantly ($p < 0.01$) negative

relationships of multi-year mean EOS with latitude, longitude and altitude. Overall, the latitudinal effect on EOS was stronger ($r_x = -0.81$) than the longitudinal effect ($r_y = -0.46$) and altitudinal effect ($r_z = -0.5$). Similar relationships were also detected in all years when yearly EOS was fitted (Table 3).

**Table 3.** Multiple regression equation parameters, evaluation indicators and partial correlation coefficients among growing season end date, latitude, longitude and altitude. $a_0$ is a constant; $a_x$, $a_y$ and $a_z$ are regression coefficients denoting latitudinal sensitivity, longitudinal sensitivity and altitudinal sensitivity, respectively; $\overline{\text{EOS}}$ is the mean growing season end date across the study area; $R^2$ is the goodness of fit of the multiple regression model; $r_x$, $r_y$ and $r_z$ are partial correlation coefficients between the growing season end date and latitude/longitude/altitude after controlling for other two geo-location indicators, respectively.

| Year | $\overline{\text{EOS}}$ (Day of the Year (DOY)) | $a_0$ | $a_x$ (d/°N) | $a_y$ (d/°E) | $a_z$ (d/100m) | RMSE (days) | $R^2$ | $r_x$ | $r_y$ | $r_z$ |
|------|------|------|------|------|------|------|------|------|------|------|
| 2000 | 298.9 | 596.05 | −2.78 | −1.30 | −1.75 | 8.8 | 0.73 *[1] | −0.72 * | −0.46 * | −0.48 * |
| 2001 | 293.7 | 590.65 | −2.41 | −1.45 | −1.48 | 8.9 | 0.71 * | −0.66* | −0.50* | −0.41* |
| 2002 | 295.9 | 605.44 | −2.90 | −1.35 | −1.86 | 9.2 | 0.72 * | −0.72 * | −0.46 * | −0.48 * |
| 2003 | 295.9 | 629.01 | −3.13 | −1.45 | −2.14 | 9.6 | 0.73 * | −0.73 * | −0.47 * | −0.52 * |
| 2004 | 299.3 | 567.87 | −3.02 | −0.99 | −1.61 | 9.4 | 0.70* | −0.72 * | −0.35 * | −0.42 * |
| 2005 | 298.5 | 569.05 | −2.97 | −1.03 | −1.54 | 8.6 | 0.73 * | −0.75* | −0.39* | −0.44* |
| 2006 | 297.1 | 559.99 | −3.83 | −0.67 | −1.29 | 9.0 | 0.78 * | −0.81 * | −0.25 * | −0.36 * |
| 2007 | 298.7 | 533.39 | −2.57 | −0.90 | −1.15 | 8.2 | 0.70* | −0.72 * | −0.36 * | −0.35 * |
| 2008 | 298.6 | 544.7 | −3.41 | −0.68 | −1.45 | 8.8 | 0.74 * | −0.78 * | −0.26 * | −0.41 * |
| 2009 | 294.6 | 623.29 | −2.36 | −1.71 | −1.66 | 8.5 | 0.75 * | −0.67 * | −0.58 * | −0.47 * |
| 2010 | 305.2 | 544.27 | −2.58 | −0.94 | −0.97 | 8.7 | 0.70 * | −0.70 * | −0.36 * | −0.29 * |
| 2011 | 300.0 | 555.19 | −3.49 | −0.72 | −1.53 | 8.4 | 0.77 * | −0.81 * | −0.29 * | −0.44 * |
| 2012 | 298.6 | 474.61 | −3.27 | −0.18 | −1.11 | 8.1 | 0.72 * | −0.80 * | −0.08 * | −0.35 * |
| Mean | 298.1 | 568.73 | −2.98 | −1.03 | −1.50 | 7.1 | 0.80 * | −0.81 * | −0.46 * | −0.50 * |

[1] * $p < 0.01$.

Moreover, both the absolute values of yearly regression equation slopes along the altitude ($a_z$) and the yearly partial correlation coefficients between EOS and the altitude ($r_z$) indicated significant decrease trends from 2000 to 2012 (Figure 6). That is, either the altitudinal sensitivity of EOS or the effect of altitude on EOS became weaker and weaker from 2000 to 2012.

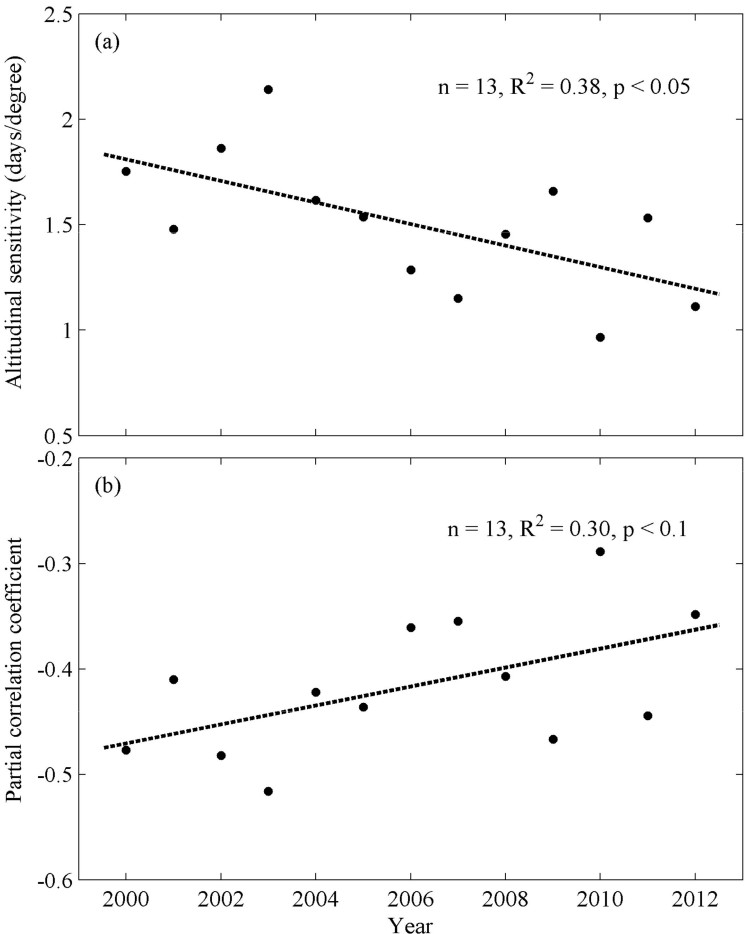

**Figure 6.** Linear trends of altitudinal sensitivity of the end date of the growing season (EOS) (**a**) and altitudinal effect on EOS (**b**) from 2000 to 2012.

### 3.3. Spatial Relationship between EOS and Climatic Factors

To investigate climatic controls of the EOS spatial pattern, we used both temperature-based and precipitation-based spatial phenology models to simulate and predict the EOS spatial series for each year. The temperature-based spatial phenology models showed a significant ($p < 0.001$) positive relationship between EOS and mean temperature during the optimum LP in each year—namely, the higher the autumn temperature at a pixel, the later the EOS date at the pixel, and vice versa. These models explained 53% to 73% of the EOS spatial variance, and yielded RMSEs of 8.7 (in 2012) to 11.3 days (in 2001) between the observed and predicted dates. Slopes of the linear models indicate that the fitted EOS varied with LP mean temperatures spatially at a rate of 2.4 days/°C (in 2010) to 3.6 days/°C (in 2003 and 2011; Figure 7). Similarly, the precipitation-based spatial phenology models also displayed a significant ($p < 0.001$) positive relationship between EOS and accumulated precipitation during the optimum LP in each year. This indicated that the more autumn accumulated precipitation at a pixel, the later EOS at the pixel, and vice versa. The precipitation-based spatial phenology models explained 10% to 46% ($p < 0.001$) of the EOS spatial variance, which were much smaller than the temperature-based spatial phenology models. In addition, the precipitation-based models yielded larger RMSEs than temperature-based spatial phenology models, which ranged from 12.7 (in 2010) to 16.7 days (in 2003; Figure S1).

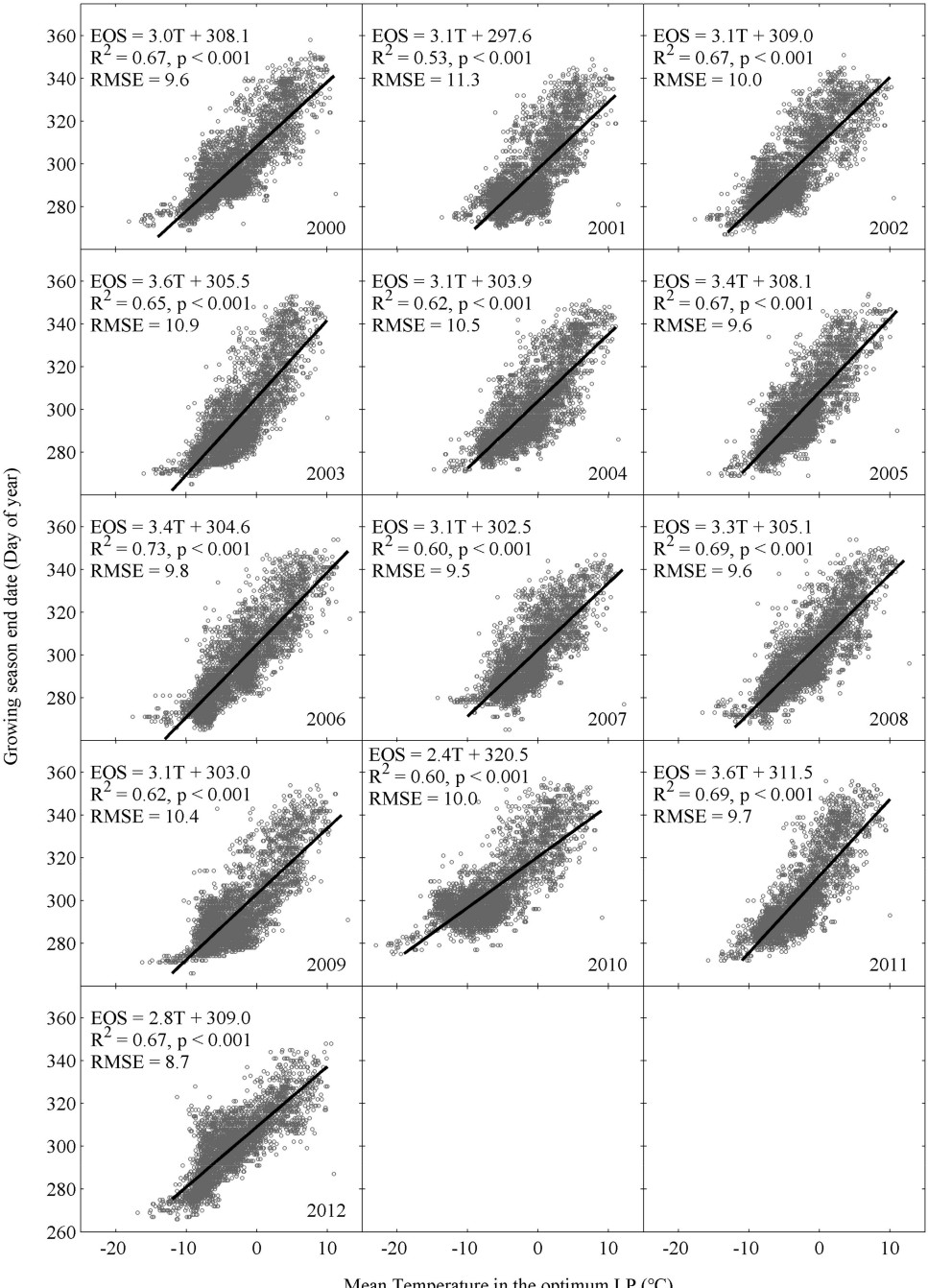

**Figure 7.** Spatial correlation and regression analyses between PSRI-derived growing season end date and mean temperature during the optimum length period (LP) in each year across the study area.

Yearly partial correlation coefficients between EOS and temperature during the optimum LP were all significant ($p < 0.001$), and ranged from 0.62 (in 2001) to 0.80 (in 2006 and 2012), while partial correlation coefficients between EOS and precipitation during the optimum LP were significant ($p < 0.001$) in 9 of 13 years, and ranged from −0.12 (in 2002) to 0.48 (in 2011). It should be noted that absolute values of partial correlation coefficients between EOS and accumulated precipitation were much smaller than those of partial correlation coefficients between EOS and the mean temperature (Table 4).

**Table 4.** Yearly partial correlation coefficients between EOS and the mean temperature/accumulated precipitation during the optimum length periods. $r_t$ denotes the partial correlation coefficient between the growing season end date and the mean temperature after eliminating the influence of accumulated precipitation; $r_p$ denotes the partial correlation coefficient between the growing season end date and accumulated precipitation after eliminating the influence of the mean temperature.

| Year | $r_t$ | $r_p$ |
|------|-------|-------|
| 2000 | 0.78 *[1] | −0.02 |
| 2001 | 0.62 * | 0.18 * |
| 2002 | 0.80 * | −0.12 * |
| 2003 | 0.76 * | 0.11 * |
| 2004 | 0.73 * | 0.10 * |
| 2005 | 0.74 * | 0.03 |
| 2006 | 0.80 * | 0.05 * |
| 2007 | 0.68 * | −0.02 |
| 2008 | 0.78 * | 0.25 * |
| 2009 | 0.76 * | −0.09 * |
| 2010 | 0.64 * | 0.17 * |
| 2011 | 0.75 * | 0.48 * |
| 2012 | 0.80 * | 0.11 * |

[1] *$p < 0.01$.

## 4. Discussion

This study assessed the ability of different remotely sensed vegetation indices in monitoring autumn phenology. Previous studies showed that different VIs are more sensitive to different aspects of forest canopies. For instance, NDVI, EVI and WDRVI are more sensitive to canopy structure and color [21,39,40], but GRVI to greenness [41], LSWI to canopy moisture content [44] and PSRI to carotenoid and mesophyll cell structure [45]. As each type of vegetation indices shows obvious seasonal characteristics that reflect local vegetation dynamics, all indices can be used for detecting autumn phenology in deciduous forests. In this study, we found that the more widely used NDVI and EVI did not perform better than other VIs in monitoring deciduous forest autumn phenology [21,23,28], and that PSRI best represented the spatial distribution of the ground-observed autumn phenology. This may be due to the fact that PSRI can capture not only the change of mesophyll cell structure but also the change of the relative proportion of carotenoid and chlorophyll in leaves, giving it greater sensitivity to leaf senescence than other VIs [45].

The multiple regression model of multi-year mean remotely sensed EOS dates showed similar latitudinal and altitudinal gradients with models of multi-year mean estimated leaf coloring dates for all three representative deciduous broadleaf species in China [46]. That is, autumn phenology occurs earlier as latitude and altitude increase. Moreover, the longitudinal gradient of EOS was consistent with that of estimated leaf coloring dates for two of three species (except *Fraxinus chinensis*), namely, autumn phenology occurs earlier as longitude increases. Although the two modeling regions did not overlap completely, consistent responses of both EOS date and leaf coloration date on geo-location indicators evidence that remotely sensed autumn phenology can reflect overall spatial variations of autumn phenology on the ground. The significantly decreasing trend of altitudinal sensitivities (in advancing days per 100 m upward) for EOS from 2000 to 2012 might be explained by larger delay rates of EOS dates at high elevations than at low elevations, considering global change. These kinds of uneven delay rates of autumn phenology at different elevations have been also detected in Slovenia for in situ European beech leaf coloration dates [47].

In this study, we employed a spatial phenology model that was originally developed using ground observations [5] to simulate multi-year mean and yearly spatial patterns of remotely sensed autumn phenology. Unlike the previously revealed spatial relationship between discrete ground autumn phenological date and mean temperature during the optimum LP [5], the remotely sensed EOS dates allowed for the characterizing of similar spatial relationships with mean temperature (delayed

autumn phenology with an increase of the optimum LP mean temperature) in a continuous manner. The consistency of temperature control on the spatial pattern of remotely sensed and ground-observed autumn phenology suggests that the spatial phenology model is applicable to both ground-observed and remotely-sensed phenological simulations. Sensitivies of remotely-sensed autumn phenology to mean temperature during optimum LPs ranged from 2.4 days/°C to 3.6 days/°C in this study (Figure 7), which is also similar to previous estimates (2.2 days/°C to 3.2 days/°C) using leaf fall dates of ground trees in northern China [5].

As anticipated, our study showed that temperature-based models can explain more spatial variations of EOS than precipitation-based models in the temperate deciduous forests. Our results further confirm that the spatial variation of remotely sensed EOS in temperate eastern China are controlled mainly by spatial variations of temperature [48], rather than precipitation. Moreover, the partial correlation coefficients between spatial EOS and spatial accumulated precipitation during the optimum LP indicate inconsistent relations (positive or negative) among years (Table 4). Thus, the underlying reasons for precipitation controls on the spatial pattern of EOS need to be further studied.

## 5. Conclusions

This study assessed and compared the reliability of six representative vegetation indices in monitoring autumn phenology, and analyzed spatial characteristics of growing season end date and its climatic controls in the temperate deciduous broadleaf forest area of northeastern China. The main conclusions are as follows:

1.  PSRI-derived EOS can more accurately capture spatial variations of ground-observed leaf fall dates of dominant trees on the multi-year average and in each year than the other five vegetation indices. The spatial variation of PSRI-derived EOS dates correlate significantly and positively with the spatial variation of ground based leaf fall dates in each year.
2.  Multi-year mean PSRI-derived EOS represent a latitudinal sensitivity of −2.98 days per degree northward, a longitudinal sensitivity of −1.03 days per degree eastward and an altitudinal sensitivity of −1.5 days per 100 m upward. The altitudinal sensitivity of EOS became weaker and weaker from 2000 to 2012, which might be attributable to larger delaying rates of EOS dates at high elevations than at low elevations under recent climate warming.
3.  Temperature-based phenological models can explain more spatial variations of EOS with less simulation errors than the precipitation-based models in the study area. Thus, the spatial variation of remotely sensed EOS are controlled mainly by spatial variations of temperature, rather than precipitation.

**Supplementary Materials:** The following are available online at http://www.mdpi.com/2072-4292/11/13/1546/s1. Figure S1: Spatial correlation and regression analyses between PSRI-derived growing season end date and accumulated precipitation during the optimum length period (LP) in each year across the study area; Table S1: Geographical coordinates and elevation of phenological stations and selected plant species at each station.

**Author Contributions:** All authors made great contributions to the work. Specific contributions include research design (W.L.; X.C.), data collection and image processing (W.L.), data analysis (W.L.), manuscript preparation and revision (W.L.; X.C.; L.L.) and discussion and suggestions (S.R.; S.Q.).

**Funding:** This research was funded by the National Natural Science Foundation of China under grant nos. 41471033, 41771049.

**Acknowledgments:** The authors thank the Meteorological Information Center of the China Meteorological Administration for providing phenological data. The China meteorological forcing dataset used in this study was developed by the Data Assimilation and Modeling Center for Tibetan Multi-Spheres, Institute of Tibetan Plateau Research, Chinese Academy of Sciences.

**Conflicts of Interest:** The authors declare no conflict of interest.

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
