# Peer review of "Geographic and Climatic Attributions of Autumn Land Surface Phenology Spatial Patterns in the Temperate Deciduous Broadleaf Forest of China"

_remotesensing, doi:10.3390/rs11131546_

Round 1

Reviewer 1 Report

The study compared the sensitivities of six vegetation indices in extracting end date of the growing season. The overall analysis was well designed presented properly. Some details need to be further addressed and below is my concern:

line 118: What was the observing scale of stations. Did it match the MODIS 500m? 

line 129: The arithmetic sum is confusing. Did you assume the precipitation in each day is an arithmetic sequence? That is not a common-sense assumption.

line 147 - 150: this description needs to be reformed. If you "replaced all VI values during the two periods", what was the point of time series analysis? I would add a figure to help explain.

line 184: Do you mean "from April to October"?

line 186-188: Are all stations covered by the 4091 pixels? If not, how many stations left?

Author Response

Comments and Suggestions for Authors

The study compared the sensitivities of six vegetation indices in extracting end date of the growing season. The overall analysis was well designed presented properly. Some details need to be further addressed and below is my concern:

Line 118: What was the observing scale of stations. Did it match the MODIS 500m? 

Response: According to the Observation Criterion of Agricultural Meteorology (In Chinese), phenological observations were normally carried out in the near of weather stations within several hundred meters and the specific spatial range of a phenological station depends on distribution of observed trees, which roughly matches spatial resolution of the MODIS data. The observed tree species were selected from the local widely distributed species, which can reflect phenological status at plant community level. Thus, we calculated the average leaf fall end date of all indicator species at each station to represent roughly the leaf fall end date at plant community levels, which should be more comparable with the remotely sensed EOS than the leaf fall end date of single species.

Line 129: The arithmetic sum is confusing. Did you assume the precipitation in each day is an arithmetic sequence? That is not a common-sense assumption.

Response: In order to avoid misunderstanding, we revised the sentence as "Daily precipitation was calculated by arithmetic sum of the eight values of accumulated precipitation every 3-hours within a day".

Line 147 - 150: this description needs to be reformed. If you "replaced all VI values during the two periods", what was the point of time series analysis? I would add a figure to help explain.

Response: We redrew the Figure 2 to illustrate MODIS data processing more clearly and revised the relevant text as follows:

Utilizing the QA and DOY flags, we identified the abnormal VI pixel values that were affected by snow, ice or clouds. Then, we replaced the abnormal VI values during start and end stages of the year using the nearest normal VI values, and other abnormal VI values with linearly interpolated VI values of the two nearest normal VI values at both sides. Moreover, we identified the date (Df) when the minimum VI (maximum PSRI) occurred in the first half reconstructed VI sequence and the date (Ds) when the minimum VI (maximum PSRI) occurred in the second half reconstructed VI sequence. All VI values from 1 January to Df and from Ds to 31 December were further replaced by the average of VI values corresponding to Df and Ds. Here, the minimum VI (maximum PSRI) value represents the initial background VI value [42]. Finally, we smoothed the VI time series with a five-point moving average filter (Figure 2).

Line 184: Do you mean "from April to October"?

Response: Yes, we used “from April to October” to replace “during April and October”.

Line 186-188: Are all stations covered by the 4091 pixels? If not, how many stations left?

Response: 15 stations are not covered by the 4091 pixels.

Reviewer 2 Report

This paper describes an interesting way to generate fall phenology in the temperate deciduous broadleaf forest of China. Overall, the topic is interesting, and the method are sound, it's a high quality paper to be published in remote sensing. There are a few things need to be addressed before publishing.

The title is "spatial simulation of ....", however, the maniscript also spend lots of work on explaining the latitudinal, longitudinal and altitudinal, temperature affect. Please try to include this in the topic to reflect the overall work.

Similar to the above comments, the author is suggested to find a way to put the work togehter, is the main object to find a way to generate fall phenology or to analyze how fall phenology relates to other factors.

Please provide more details on the phenological stations, such as their representativeness, the size and such, to make sure the validation process can be trusted.

The section of 2.5 Spatial phenology model based on climatic factors is a little bit confusion for me. If this model is used for prediction, then there should be a uniform one in this region, while figure 6 showed one for each year.

There are few equations for modeling EOS, please explain their relation

please double check the projection in figure 1 for the region and China used the same projection.

Also please check the way to cite tables and figures in supplements, should it be S1 or A1?

Author Response

This paper describes an interesting way to generate fall phenology in the temperate deciduous broadleaf forest of China. Overall, the topic is interesting, and the method are sound, it's a high quality paper to be published in remote sensing. There are a few things need to be addressed before publishing.

The title is "spatial simulation of ....", however, the manuscript also spend lots of work on explaining the latitudinal, longitudinal and altitudinal, temperature affect. Please try to include this in the topic to reflect the overall work.

Response: Many thanks for your comments. We changed the topic to “Geographic and climatic attributions of autumn land surface phenology spatial patterns in the Temperate Deciduous Broadleaf Forest of China”.

Similar to the above comments, the author is suggested to find a way to put the work together, is the main object to find a way to generate fall phenology or to analyze how fall phenology relates to other factors.

Response: According to your suggestions, we put multiple regression model among EOS date and geo-location indicators and spatial phenology model based on climatic factors together into section 2.4 Spatial phenology models based on geo-location indicators and climatic factors. In addition, we also revised the topics of section 3.2 and 3.3 as follows:

3.2 Spatial patterns of EOS and their relation to geo-location factors”

“3.3 Spatial relationship between EOS and climatic factors”

Please provide more details on the phenological stations, such as their representativeness, the size and such, to make sure the validation process can be trusted.

Response: According to the Observation Criterion of Agricultural Meteorology (In Chinese), phenological observations were normally carried out in the near of weather stations within several hundred meters and the specific spatial range of a phenological station depends on distribution of observed trees, which roughly matches spatial resolution of the MODIS data. The observed tree species were selected from the local widely distributed species, which can reflect phenological status at plant community level. Thus, we calculated the average leaf fall end date of all indicator species at each station to represent roughly the leaf fall end date at plant community levels, which should be more comparable with the remotely sensed EOS than the leaf fall end date of single species.

The section of 2.5: Spatial phenology model based on climatic factors is a little bit confusion for me. If this model is used for prediction, then there should be a uniform one in this region, while figure 6 showed one for each year.

Response: The basic hypothesis for plant phenology spatial series modeling is that the station-to-station variation of a phenological event occurrence date over a region is mainly influenced by station-to-station variation of mean temperature/accumulated precipitation within a particular length period during and before its occurrence over the region. Thus, a spatial phenology model can simulate plant phenology spatial series based on temperature and precipitation spatial series in the research region in each year, namely, there is a uniform model in the research region in each year.

There are few equations for modeling EOS, please explain their relation.

Response: Eq.5 shows that the entire length period (LP) during which climatic factors influence EOS consists of two sub-periods: one is the basic length period (bLP) between the earliest and latest dates of EOS spatial series across all the pixels in a given year, the other is the moving length period (mLP) prior to the earliest date of the EOS spatial series. The optimum autumn LP in the research region for each year is determined based on the highest correlation between the EOS spatial series and the corresponding mean temperature/accumulated precipitation spatial series. Eq.6 is the linear regression model between mean temperature/accumulated precipitation during the optimum length period and EOS.

Please double check the projection in figure 1 for the region and China used the same projection.

Response: We revised the projection for China in figure 1, so that it is the same with the projection for the research region.

Also please check the way to cite tables and figures in supplements, should it be S1 or A1?

Response: The table in supplementary materials should be S1. We have corrected it.

Reviewer 3 Report

This manuscript presents the derivation of a spatial explicit end date of growing season (EOS) variable investigating different vegetation index products derived from MODIS data for a part of China including topographic varying terrain. The satellite derived EOS product was then evaluated using many locations of ground-observed leaf fall data at 27 stations using 13 years of data.

Overall, it is an interesting and relevant paper that covers an interesting topic that nicely fits into the scope of RS. The paper is well written, based on sufficient large datasets, and the figures and tables are nicely presenting the research results. I would suggest to change the title since it is not a simulation: “A remotely sensed spatial explicit leaf fall data sets in a temperate deciduous broadleaf forest in China” or something similar. The filtering procedure of the VI data is not clear and needs more explanation, as well as some other issues should be dealt with (see infra). Since the temperature and rainfall based spatial phenology models are so important for this research, much more detail must be given to this models in this manuscript.

L19: “dependency on”;

L22: “a 1 K (Kelvin) spatial shift”;

L27: ” … might lead to …”;

ML36: “to temporal variationS of …”;

L93: insert space between 1456.54 and mm;

L126: three hours, or 3-hours;

L128-129: 8 should be eight;

L132: inert space between 500 and m;

Table 1. I suggest to include the abbreviations of the different vegetation indices at the bottom of the table.

L144-152: I do not understand the replacing of the VI values during the two periods to its mean VI minimum. What is a mean minimum? Does it mean that you define the periods where all the VI values are set to some minimum VI value? Then a 3-datapoint moving average filter is applied? Perhaps it would be better to extend figure 2 with different figure panels showing the different steps in each panel starting from the “raw” VI data, then which VI values are considered for the minimum, which VI values are replaced, etc For now, I do not completely understand the followed procedure.

L154: remove “shown”;

Figure 2. See before mentioned comment. In the caption: “… determining THE end of the …”;

L172: THE root mean …;

L175: “… was made both for individual years as well as for a multi-year average.”;

L186: What is an intersect overlay analysis? Please elaborate;

L188: 4091 common pixels: Do the authors mean 4091 pixels with 2000-2012 time series (each of the 4091 pixel is a time series?)? Or 4091 both is space as in time? Please Clarify;

L189: “To examine THE attribution of …”;

L198: dependency;

L202-207: Include a scheme to better explain the procedure of the LP computations;  

L208: A temperature –based spatial phenology model? What kind of model is this? Is it a statistical model? Please elaborate; It is a crucial part of the paper.

L227: ”We used THE spatial RMSE to ….”;

L229: six;

L235: “To evaluate THE spatial variation …”;

L237: “… and for each year …”;

Figure 4: include numbering of the panels;

L254: “is demonstrated”, not evidenced;

L261: Spatial patternS;

L274: multi-year;

Table 3 should be completely on 1 page, not broken;

Figure 6. Caption: I assume that t is the temperature in the linear equation? Using T should be better;

Table 4: caption: What do the authors mean with “controlling”? Please elaborate.

L350: declining trend;

L351: might be explained by;

L351: delay rates;

L352: considering the global change;

L353: delay rates;

L369: further confirm that the …;,

L371: Moreover, THE partial …;

L373: Thus, THE underlying … on THE spatial pattern …;

L387: … which MIGHT be attributed …;

L389: temperature-based phenological models …

Author Response

Comments and Suggestions for Authors

This manuscript presents the derivation of a spatial explicit end date of growing season (EOS) variable investigating different vegetation index products derived from MODIS data for a part of China including topographic varying terrain. The satellite derived EOS product was then evaluated using many locations of ground-observed leaf fall data at 27 stations using 13 years of data.

  Overall, it is an interesting and relevant paper that covers an interesting topic that nicely fits into the scope of RS. The paper is well written, based on sufficient large datasets, and the figures and tables are nicely presenting the research results. I would suggest to change the title since it is not a simulation: “A remotely sensed spatial explicit leaf fall data sets in a temperate deciduous broadleaf forest in China” or something similar. The filtering procedure of the VI data is not clear and needs more explanation, as well as some other issues should be dealt with (see infra). Since the temperature and rainfall based spatial phenology models are so important for this research, much more detail must be given to this models in this manuscript.

Response to the point 1: Many thanks for your comments. Considering suggestions from you and other reviewers, we changed the title to “Geographic and Climatic Attributions of Autumn Land Surface Phenology Spatial Patterns in the Temperate Deciduous Broadleaf Forest of China”, which may more correctly reflect contents and main results of this article.

Response to the point 2: We rephrased the paragraph on the filtering procedure of the VI data in the section 2.3 (MODIS data and processing) as follows:

Utilizing the QA and DOY flags, we identified the abnormal VI pixel values that were affected by snow, ice or clouds. Then, we replaced the abnormal VI values during start and end stages of the year using the nearest normal VI values, and other abnormal VI values with linearly interpolated VI values of the two nearest normal VI values at both sides. Moreover, we identified the date (Df) when the minimum VI (maximum PSRI) occurred in the first half reconstructed VI sequence and the date (Ds) when the minimum VI (maximum PSRI) occurred in the second half reconstructed VI sequence. All VI values from 1 January to Df and from Ds to 31 December were further replaced by the average of VI values corresponding to Df and Ds. Here, the minimum VI (maximum PSRI) value represents the initial background VI value [42]. Finally, we smoothed the VI time series with a five-point moving average filter (Figure 2).

Response to the point 3: We provided much more details of spatial phenology model based on climatic factors through extensive revision (See below).

“We used mean air temperature and accumulated precipitation spatial series during the optimum length period to correlate with EOS spatial series. The basic hypothesis is that the station-to-station variation of a phenological event occurrence date over an area is mainly influenced by station-to-station variation of climatic factors within a particular length period (LP) of days during and before its occurrence over the area [5]. The LP in this study is determined as the time periods during which the pixel-to-pixel variation in daily mean temperature or accumulated precipitation affects the pixel-to-pixel variation of EOS most remarkably in a year. The computation process was implemented through following steps. First, we defined the number of days between the earliest and latest dates of EOS spatial series across all the pixels in a given year as the basic length period (bLP). Then, we calculated backwards (to earlier time direction) the mean air temperature and accumulated precipitation spatial series across all the pixels in the year during bLP+mLP. Here, mLP was a moving length period prior to the earliest date of EOS spatial series, which moved with a step length of 1 day, namely, during bLP+1 day, bLP+2 day, bLP+3 day, etc. The maximum mLP was set to 60 days because mean temperature and accumulated precipitation prior to this period were usually assumed to have limited effects on spatial variation of occurrence dates of a phenological event. Thus, the LP is defined as follows [Eq. (5)]:

  LP=bLP+mLP ,

(5)

Further, we calculated correlation coefficients between EOS spatial series and mean temperature/accumulated precipitation spatial series during different testing LPs across all the pixels in a specific year. Finally, we obtained the optimum LP based on the highest correlation between the EOS spatial series and the corresponding mean temperature/accumulated precipitation spatial series. The temperature or precipitation within the optimum LP across all the pixels in the year was thought as the major influence factor of EOS spatial variation. Hence, the optimal temperature/precipitation-based spatial phenology model for each year can be created as [Eq. (6)]:

EOS=a+b×X,

(6)

where X is the mean temperature or accumulated precipitation during the optimum LP; a and b are intercept and slope of the equation, respectively. To evaluate the relative importance of temperature and precipitation on controlling EOS spatial patterns, we further executed a partial correlation analysis between EOS spatial series and temperature/precipitation spatial series during the optimum LP. The significance levels of partial correlation were evaluated based on two-tailed significance tests.”

L19: “dependency on”

Response: Thanks. We have replaced “dependence” with “dependency”.

L22: “a 1 K (Kelvin) spatial shift”;

Response: Thanks. We have replaced “°C” with “K (Kelvin)”.

L27: ” … might lead to …”;

Response: Revision has been done as suggested.

L36: “to temporal variationS of …”;

Response: Revision has been done as suggested.

L93: insert space between 1456.54 and mm;

Response: Revision has been done as suggested.

L126: three hours, or 3-hours;

Response: Revision has been done as suggested.

L128-129: 8 should be eight;

Response: Thanks. We have replaced “8” with “eight”.

L132: inert space between 500 and m;

Response: Revision has been done as suggested.

Table 1. I suggest to include the abbreviations of the different vegetation indices at the bottom of the table.

Response: Revision has been done as suggested.

L144-152: I do not understand the replacing of the VI values during the two periods to its mean VI minimum. What is a mean minimum? Does it mean that you define the periods where all the VI values are set to some minimum VI value? Then a 3-datapoint moving average filter is applied? Perhaps it would be better to extend figure 2 with different figure panels showing the different steps in each panel starting from the “raw” VI data, then which VI values are considered for the minimum, which VI values are replaced, etc. For now, I do not completely understand the followed procedure.

Response: Please see the response to your major comment point 2.  

L154: remove “shown”;

Response: Revision has been done as suggested.

Figure 2. See before mentioned comment. In the caption: “… determining THE end of the …”;

Response: Revision has been done as suggested.

L172: THE root mean …;

Response: Revision has been done as suggested.

L175: “… was made both for individual years as well as for a multi-year average.”;

Response: Revision has been done as suggested.

L186: What is an intersect overlay analysis? Please elaborate;

Response: We rephrased the sentences as follows:

Due to the removal of abnormal pixels, the total numbers of pixels in TDBF are different in different years. To ensure the same geographic range among different years, we extracted the common pixels from 2000 to 2012 and obtained 4091 pixels for further study.

L188: 4091 common pixels: Do the authors mean 4091 pixels with 2000-2012 time series (each of the 4091 pixel is a time series?)? Or 4091 both is space as in time? Please Clarify;

Response: Yes, each of the 4091 pixels is a time series from 2000 to 2012.

L189: “To examine THE attribution of …”;

Response: Revision has been done as suggested.

L198: dependency;

Response: We have replaced “dependence” with “dependency”.

L202-207: Include a scheme to better explain the procedure of the LP computations;

Response: Please see the response to your major comment point 3.

L208: A temperature –based spatial phenology model? What kind of model is this? Is it a statistical model? Please elaborate; It is a crucial part of the paper.

Response: Yes, the temperature-based spatial phenology model is a statistical model. We provided much more details of spatial phenology model based on climatic factors through extensive revision. Please see the response to your major comment point 3.

L227: ”We used THE spatial RMSE to ….”;

Response: Revision has been done as suggested.

L229: six;

Response: Revision has been done as suggested.

L235: “To evaluate THE spatial variation …”;

Response: Revision has been done as suggested.

L237: “… and for each year …”;

Response: Revision has been done as suggested.

Figure 4: include numbering of the panels;

Response: Thanks. We added the numbering of the panels in Figure 4.

L254: “is demonstrated”, not evidenced;

Response: Thanks. We have replaced evidenced with demonstrated.

L261: Spatial patternS;

Response: Revision has been done as suggested.

L274: multi-year;

Response: Revision has been done as suggested.

Table 3 should be completely on 1 page, not broken;

Response: Thanks. We have rearranged Table 3 to be completely on 1 page.

Figure 6. Caption: I assume that t is the temperature in the linear equation? Using T should be better;

Response: Thanks for your advice. We replaced “t” with “T” in the linear equations.

Table 4: caption: What do the authors mean with “controlling”? Please elaborate.

Response: To investigate the individual effect of temperature and precipitation on EOS, we calculated partial correlation coefficients between temperature/precipitation and EOS after eliminating the influence of precipitation/temperature. We have replaced “controlling” with “eliminating the influence of”.

L350: declining trend;

Response: Revision has been done as suggested.

L351: might be explained by;

Response: Revision has been done as suggested.

L351: delay rates;

Response: Revision has been done as suggested.

L352: considering the global change;

Response: Revision has been done as suggested.

L353: delay rates;

Response: Revision has been done as suggested.

L369: further confirm that the …;,

Response: Revision has been done as suggested.

L371: Moreover, THE partial …;

Response: Revision has been done as suggested.

L373: Thus, THE underlying … on THE spatial pattern …;

Response: Revision has been done as suggested.

L387: … which MIGHT be attributed …;

Response: Revision has been done as suggested.

L389: temperature-based phenological models …

Response: Revision has been done as suggested.

Reviewer 4 Report

This paper builds on previous work by combining two areas of previous study. Specifically, work that examined the impact of temperature controls on ground observations on autumn phenology, as well as work in the remote sensing literature that has identified the PSRI index as particularly suitable for detection of the end of the growing season. This paper explores the timing of the end of the growing season, which has generally been more difficult to model than the start of the growing season and assesses the variation in the timing of the end of the season due to temperature, altitude and precipitation. The paper is well-written and well-structured.

Requested changes:

The main changes are slight changes to the text. However, S1 should be moved from the supplementary material to the main body of the text as it is referred to in the discussion and in the abstract and the trend is not obvious by just looking at the numbers in Table 3.

Title: I would suggest changing the title slightly to ‘Factors influencing autumn phenology in the temperate deciduous broadleaf forest of China’

Line 23: rephrase slightly to ‘The models explain between 54% and 73% of the variance in the EOS timing.’

Line 33: change ‘consequences to’ to ‘impacts on’

Line 34: change ‘showed’ to ‘show’

Line 36-line 38: needs rephrasing slightly i would suggest: ‘current research is focused mainly on understanding the relationship between autumn phenology and climate [8,9], which has shown the importance of autumn temperatures in determining the timing of leaf senescence [10,11]. Whereas, the spatial variation of autumn phenology and its climatic controls have received much less attention.’

Line 47: change ‘and’ to ‘or’

Line 76: please avoid the use of ‘ground truth’ as a phrase, ‘ground reference’ is a better term.

Line 84: i’m not sure about the use of ‘evident’ here. I think, ‘distinct’ or ‘characteristic’ might be better.

Line 115: change ‘data’ to ‘date’

Line 136-139: I would define the NDVI and PSRI here (as well as the introduction) in case readers have skipped the introduction and the PSRI is not as well known as the NDVI.

Line 147-150: I found this confusing the first time I read it. I think it needs an extra line in above to explain what you are doing and why i.e. the data outside the growing season are noisy, so to fit a good curve through the points the data are ‘cleaned’. This is done by identifying a fixed value for points outside the growing season. These points are identified by….

Line 161: change to ‘and for each year, as they are the dates of the last local…’

Line 162: change ‘fulfilled’ to ‘conducted’

Line 181: change ‘pixel by pixel over’ to ‘for each pixel’

Line 186-188: rephrase slightly. Suggest ‘We extracted the pixels with both climate data and a valid EOS estimate, this resulted in 4091 pixels for further study.’

Line 333: change ‘to’ to ‘that’

Line 336: change to ‘that PSRI best represented the spatial distribution of the ground-observed autumn phenology.’

Line 339: change ‘hence…’ to ‘giving it greater sensitivity to leaf senescence that other VIs [44].’

Lines 341-line 344 - it’s not clear whether this is referring to the work in reference 45 or the work presented here, so the text needs clarifying.

Line 350: change ‘decline’ to ‘decreasing’

Author Response

Comments and Suggestions for Authors

This paper builds on previous work by combining two areas of previous study. Specifically, work that examined the impact of temperature controls on ground observations on autumn phenology, as well as work in the remote sensing literature that has identified the PSRI index as particularly suitable for detection of the end of the growing season. This paper explores the timing of the end of the growing season, which has generally been more difficult to model than the start of the growing season and assesses the variation in the timing of the end of the season due to temperature, altitude and precipitation. The paper is well-written and well-structured.

Requested changes:

The main changes are slight changes to the text. However, S1 should be moved from the supplementary material to the main body of the text as it is referred to in the discussion and in the abstract and the trend is not obvious by just looking at the numbers in Table 3.

Response: Thanks for your suggestion. We have moved Figure S1 from the supplementary material to the main body of the text, namely, Figure 6.

Title: I would suggest changing the title slightly to ‘Factors influencing autumn phenology in the temperate deciduous broadleaf forest of China’

Response: Considering suggestions from you and other reviewers, we changed the title to “Geographic and Climatic Attributions of Autumn Land Surface Phenology Spatial Patterns in the Temperate Deciduous Broadleaf Forest of China”, which may more correctly reflect contents and main results of this article.

Line 23: rephrase slightly to ‘The models explain between 54% and 73% of the variance in the EOS timing.’

Response: Revision has been done as suggested.

Line 33: change ‘consequences to’ to ‘impacts on’

Response: Revision has been done as suggested.

Line 34: change ‘showed’ to ‘show’

Response: Revision has been done as suggested.

Line 36-line 38: needs rephrasing slightly i would suggest: ‘current research is focused mainly on understanding the relationship between autumn phenology and climate [8,9], which has shown the importance of autumn temperatures in determining the timing of leaf senescence [10,11]. Whereas, the spatial variation of autumn phenology and its climatic controls have received much less attention.’

Response: Revision has been done as suggested.

Line 47: change ‘and’ to ‘or’

Response: Revision has been done as suggested.

Line 76: please avoid the use of ‘ground truth’ as a phrase, ‘ground reference’ is a better term.

Response: Revision has been done as suggested.

Line 84: i’m not sure about the use of ‘evident’ here. I think, ‘distinct’ or ‘characteristic’ might be better.

Response: We have replaced “evident” with “distinct”.

Line 115: change ‘data’ to ‘date’

Response: We rephrased this sentence as follows:

We utilized ground-observed 13-year continuous record of leaf fall end date of each species and MODIS data from 2000 to 2012.

Line 136-139: I would define the NDVI and PSRI here (as well as the introduction) in case readers have skipped the introduction and the PSRI is not as well known as the NDVI.

Response: Revision has been done as suggested.

Line 147-150: I found this confusing the first time I read it. I think it needs an extra line in above to explain what you are doing and why i.e. the data outside the growing season are noisy, so to fit a good curve through the points the data are ‘cleaned’. This is done by identifying a fixed value for points outside the growing season. These points are identified by….

Response: We rephrased this paragraph as follows:

Utilizing the QA and DOY flags, we identified the abnormal VI pixel values that were affected by snow, ice or clouds. Then, we replaced the abnormal VI values during start and end stages of the year using the nearest normal VI values, and other abnormal VI values with linearly interpolated VI values of the two nearest normal VI values at both sides. Moreover, we identified the date (Df) when the minimum VI (maximum PSRI) occurred in the first half reconstructed VI sequence and the date (Ds) when the minimum VI (maximum PSRI) occurred in the second half reconstructed VI sequence. All VI values from 1 January to Df and from Ds to 31 December were further replaced by the average of VI values corresponding to Df and Ds. Here, the minimum VI (maximum PSRI) value represents the initial background VI value [42]. Finally, we smoothed the VI time series with a five-point moving average filter (Figure 2).

Line 161: change to ‘and for each year, as they are the dates of the last local…’

Response: Revision has been done as suggested.

Line 162: change ‘fulfilled’ to ‘conducted’

Response: Revision has been done as suggested.

Line 181: change ‘pixel by pixel over’ to ‘for each pixel’

Response: Revision has been done as suggested.

Line 186-188: rephrase slightly. Suggest ‘We extracted the pixels with both climate data and a valid EOS estimate, this resulted in 4091 pixels for further study.’

Response: We rephrased the sentences as follows:

Due to the removal of abnormal pixels, the total numbers of pixels in TDBF are different in different years. To ensure the same geographic range among different years, we extracted the common pixels from 2000 to 2012 and obtained 4091 pixels for further study.

Line 333: change ‘to’ to ‘that’

Response: Revision has been done as suggested.

Line 336: change to ‘that PSRI best represented the spatial distribution of the ground-observed autumn phenology.’

Response: Revision has been done as suggested.

Line 339: change ‘hence…’ to ‘giving it greater sensitivity to leaf senescence that other VIs [44].’

Response: Revision has been done as suggested.

Lines 341-line 344 - it’s not clear whether this is referring to the work in reference 45 or the work presented here, so the text needs clarifying.

Response: This is referring to the work in reference 45, namely, the reference 46 in the revised manuscript. Thus, we cited the reference 46 directly following this sentence.

Line 350: change ‘decline’ to ‘decreasing’

Response: Revision has been done as suggested.

Reviewer 5 Report

I have found that the paper is well organized and scientifically solid to be considered for publication.

My specific comments and recommendations are as follows

L118-120 The leaf fall end date (LF) represents “the ground truth”, a crucial point in the validation process of the spatial phenology model. For 10 of the 27 phenological stations, plant communities involve different species. The calculation of LF as the simple average of the date recorded, at least, on three individual plants for each species, is probably to be considered as a weighted average taking into account the composition of plant community. Is it correct?

L134 The bands 6 and 7, computed from the MODIS level 1B are not applied in the VI calculation. Is it useful to report them in the text?

L147-150 The explanation of the method to apply VI’s values for the periods with no useful data, should be better written.

L166 Delete “Black”. It should be “White”.

L182-188 This is not statistical analysis but a data treatment methodology. Probably it should be inserted in 2.3. section.

L184 The “original EOS” should be considered “validated EOS”. Is it correct?

L183 As described for VI’s replacement methods (Line 147-150), different behaviour for PSRI should be emphasized (minimum values).

L 189-199 The study of the influence of geolocation factors on EOS spatial patterns in this paper produced interesting results useful for spatial phenology models. Probably this section should be added to 2.5.paragraph renamed as “Spatial phenology model based on geolocation and climatic factors”.

L373-374 The relatively low effect of soil water availabitilty on forest could partially explain the low sensitivity observed

Figura 4 Add letter for each subfigure as described in L241-242

Supplementary material Table A1: are the geolocation data are refered to the centroid? What’s the total area for each station?

Author Response

Comments and Suggestions for Authors

I have found that the paper is well organized and scientifically solid to be considered for publication.

My specific comments and recommendations are as follows:

L118-120 The leaf fall end date (LF) represents “the ground truth”, a crucial point in the validation process of the spatial phenology model. For 10 of the 27 phenological stations, plant communities involve different species. The calculation of LF as the simple average of the date recorded, at least, on three individual plants for each species, is probably to be considered as a weighted average taking into account the composition of plant community. Is it correct?

Response: Yes, we agree that a weighted average should be considered. Taking into account the composition of plant community will be helpful for effective validation. However, as the leaf fall end date is identified as the day when more than 90% of the leaves fall off in more than half of the observed trees (at least three individual plants), we cannot further calculate the weighted average date from the raw dataset.

L134 The bands 6 and 7, computed from the MODIS level 1B are not applied in the VI calculation. Is it useful to report them in the text?

Response: Thanks for your comments. We rephrased the relevant sentences as follows:

“Surface reflectance was computed from seven MODIS Level 1B bands (band 1 to 7 centered at 648 nm, 858 nm, 470 nm, 555 nm, 1240 nm, 1640 nm, and 2130 nm, respectively) after removal of the atmospheric effect [36], in which the surface reflectance data from bands 1-5 were used to calculate the six types of vegetation indices,…”

L147-150 The explanation of the method to apply VI’s values for the periods with no useful data, should be better written.

Response: We rephrased this paragraph as follows:

Utilizing the QA and DOY flags, we identified the abnormal VI pixel values that were affected by snow, ice or clouds. Then, we replaced the abnormal VI values during start and end stages of the year using the nearest normal VI values, and other abnormal VI values with linearly interpolated VI values of the two nearest normal VI values at both sides. Moreover, we identified the date (Df) when the minimum VI (maximum PSRI) occurred in the first half reconstructed VI sequence and the date (Ds) when the minimum VI (maximum PSRI) occurred in the second half reconstructed VI sequence. All VI values from 1 January to Df and from Ds to 31 December were further replaced by the average of VI values corresponding to Df and Ds. Here, the minimum VI (maximum PSRI) value represent the initial background VI value [42]. Finally, we smoothed the VI time series with a five-point moving average filter (Figure 2).

L166 Delete “Black”. It should be “White”.

Response: We revised the figure caption by using circle instead of black circle.

L182-188 This is not statistical analysis but a data treatment methodology. Probably it should be inserted in 2.3. section.

Response: We agree with you and have moved this part to Section 2.3.

L184 The “original EOS” should be considered “validated EOS”. Is it correct?

Response: The “original EOS” means the EOS extracted from the best performing VI at the spatial resolution of 500 m. In order to avoid misunderstanding, we use the “satellite-derived EOS” to replace the “original EOS”.

L183 As described for VI’s replacement methods (Line 147-150), different behaviour for PSRI should be emphasized (minimum values).

Response: Revision has been done as suggested.

L 189-199 The study of the influence of geolocation factors on EOS spatial patterns in this paper produced interesting results useful for spatial phenology models. Probably this section should be added to 2.5.paragraph renamed as “Spatial phenology model based on geolocation and climatic factors”.

Response: Thanks for your suggestion. We have moved this part to the section “Spatial phenology models based on geo-location indicators and climatic factors”.

L373-374 The relatively low effect of soil water availabitilty on forest could partially explain the low sensitivity observed

Response: As soil water data are not available in the study area, we cannot prove your viewpoint. Therefore, we didn’t explain the possible attribution of inconsistent spatial correlations between EOS and accumulated precipitation among different years.

Figure 4 Add letter for each subfigure as described in L241-242

Response: Thanks. We added the numbering of the panels in Figure 4.

Supplementary material Table A1: are the geolocation data are refered to the centroid? What’s the total area for each station?

Response: According to the Observation Criterion of Agricultural Meteorology (In Chinese), phenological observations were normally carried out in the near of weather stations within several hundred meters and the specific spatial range of a phenological station depends on distribution of observed trees. The geo-location data refer to the centroid of phenological stations.

Round 2

Reviewer 3 Report

L23: remove (Kelvin), just K;

L161: 5-point or five point;

L180-181: include in caption that the smoothing uses the 5-point moving average;

L392: should be considering global change, so without the THE;

Author Response

Comments and Suggestions for Authors

L23: remove (Kelvin), just K;

Response: Revision has been done as suggested.

L161: 5-point or five point;

Response: Revision has been done as suggested.

L180-181: include in caption that the smoothing uses the 5-point moving average;

Response: Revision has been done as suggested.

L392: should be considering global change, so without the THE;

Response: Revision has been done as suggested. The line should be L375.